# Calcium-Based Binders in Concrete or Soil Stabilization: Challenges, Problems, and Calcined Clay as Partial Replacement to Produce Low-Carbon Cement

**DOI:** 10.3390/ma16052020

**Published:** 2023-02-28

**Authors:** Angham Ali Mohammed, Haslinda Nahazanan, Noor Azline Mohd Nasir, Ghasan Fahim Huseien, Ahmed Hassan Saad

**Affiliations:** 1Department of Civil Engineering, Faculty of Engineering, University of Putra Malaysia, Seri Kembangan 43400, Selangor, Malaysia; 2Department of the Build Environment, School of Design and Environment, National University of Singapore, Singapore 117566, Singapore

**Keywords:** calcium-based binders, cement, lime, problems of cement and lime, sulfate attack, CO_2_ emission, alternative materials (calcined clay)

## Abstract

Calcium-based binders, such as ordinary Portland cement (OPC) and lime (CaO), are the most common artificial cementitious materials used worldwide for concrete and soil improvement. However, using cement and lime has become one of the main concerns for engineers because they negatively affect the environment and economy, prompting research into alternative materials. The energy consumption involved in producing cementitious materials is high, and the subsequent CO_2_ emissions account for 8% of the total CO_2_ emissions. In recent years, an investigation into cement concrete’s sustainable and low-carbon characteristics has become the industry’s focus, achieved by using supplementary cementitious materials. This paper aims to review the problems and challenges encountered when using cement and lime. Calcined clay (natural pozzolana) has been used as a possible supplement or partial substitute to produce low-carbon cement or lime from 2012–2022. These materials can improve the concrete mixture’s performance, durability, and sustainability. Calcined clay has been utilized widely in concrete mixtures because it produces a low-carbon cement-based material. Owing to the large amount of calcined clay used, the clinker content of cement can be lowered by as much as 50% compared with traditional OPC. It helps conserve the limestone resources used in cement manufacture and helps reduce the carbon footprint associated with the cement industry. Its application is gradually growing in places such as Latin America and South Asia.

## 1. Introduction

Soil stabilization is a very useful technique for civil engineering work. Soil stabilization is the modification of one or more soil characteristics through chemical or mechanical methods to generate better soil material containing the desired engineering properties. Soils may also be stabilized to enhance durability and strength or to limit dust production and erosion.

Regardless of the aim for the stabilization, the desired result is the formation of a soil system or soil material that will remain and sustain in place under the design use conditions for the design life of the construction [1]. Traditional stabilization methods include the use of cement, lime, and waste materials. By enhancing various soil engineering properties, these stabilization methods generate an improved construction material [2]. Limestone is one of the most common crushed rocks and is an important part of building materials such as cement, lime, and building stones [3]. Using hydraulic binders in soil improvement is a widespread practice in foundation work [4].

Lime additive provides better index properties, increases the unconfined compression strength, increases the California bearing ratio value, reduces dispersity with an increasing lime quantity and curing period, and decreases the hydraulic conductivity [4,5,6,7]. Adding lime to soil also decreases the liquid limit, plasticity, and maximum dry density while slightly increasing the plastic limit and optimum moisture content [5]. Researchers [8] also showed that as the proportion of lime increased, the density decreased and the optimum moisture content increased. For lime-stabilized soils, clay minerals are considered the primary targets of chemical attack, and the cementitious products (C-A-S-H and C-S-H), which affect pore size distribution and the gains in strength, are related to the progressive creation of these new phases [9,10]. For the cement additive, the cementation effect improved the shear strength, stiffness, and significantly increased the pre-consolidation pressure of the soft soil and decreased the compressibility parameters and settlement of the treated soft soil [11,12,13]. The cement component material causes a decrease in the sensitivity to the moisture of the expansive soil, and the cement-induced hydration reaction decreases swelling and shrinkage while raising unconfined compressive strength and resilient modulus but lowering strain at failure [14].

Regardless of the improvements in the soil characteristics achieved by using cement or lime, these calcium-based binders have shown some shortcomings specific to the environment. This is a problem because, at present, the global trend from different approaches is to reduce environmental pollution, as shown in many studies [15,16]. For example, traditional binder cement generates as much as 5% of artificial CO_2_ emissions [17,18]. The OPC industry is under tight security due to the emission of huge volumes of CO_2_ in manufacturing clinker. In terms of global anthropogenic CO_2_ emissions, the cement sector may be responsible for as much as 5%, according to some estimations [19]. Every ton of cement produces one-quarter ton equivalent of CO_2_ released [20]. Cement manufacturing is recognized as being responsible for around 7.4% of the world’s carbon dioxide emissions (2.9 Gt in 2016) [21]. In 2016, the global cement production was roughly 4.65 Gt, to which China contributing 52% and the rest of Asia contributed 28.5%. Europe (The European Cement Association (CEMBUREAU) members) only makes 5.3% of the world’s cement. The cumulative global process emissions of carbon dioxide from 1928 and 2018 were 38.3 ± 2.4 Gt CO_2_, 71% of which have occurred since 1990 [22]. Therefore, China is considered to be the largest producer and consumer of cement, worldwide [23]. The cement industry produces approximately one-eighth of China’s total anthropogenic carbon dioxide emissions, and emissions of CO_2_ increased 5.8 times in 2008 [24]. Fossil fuel production and clinkering jointly account for more than 70% of CO_2_ equivalent [25]. The emissions in the OPC sector are 0.662 t CO_2_/ton of produced cement [26]. The concentrations of carbon dioxide emissions in Nigeria are in the range of 2440–2600 mg/m^3^ [27].

In 2015, the gross CO_2_ emissions from cement were approximately 840 kg/ton (China–Korea–Japan), 863 kg/ton (Central America), 845 kg/ton (Middle East), 880 kg/ton (North America), 830 kg/ton (Africa), and 825 kg/ton (EU) [28].

The coefficient of CO_2_ emissions for lime was proposed to be 0.12 Mg C per Mg for CaCO_3_ [20], which explained that 100% of carbon in CaCO_3_, is ultimately released to the atmosphere as CO_2_. Major greenhouse emissions are implicated in global warming [29]. Another major problem that has been challenging for underground construction is corrosion due to sulfate attacks. In light of recent heaving and premature pavement failures in cement and lime-treated subgrades containing sulfates, the efficacy of calcium-based stabilization has been called into question. The stabilizers based on calcium react with soil sulfates and alumina to produce ettringite, a mineral that increases the expansive properties of sulfate-rich soils. In addition to this problem, the high energy demand and the high cost of cement used as a binder in mortar production have led to a search for alternatives to use as a partial replacement for lime and cement in concrete and soil stabilization. One of these alternatives is the clay slag binder activated with sodium carbonate, sodium silicate, and calcium hydroxide solutions, which has shown improved strength and durability [30]. Therefore, this paper primarily aims to review the major problems of using calcium-based binders in soft soil stabilization, to propose solutions to the problems by using calcined clay as a partial replacement, and to compare the efficiency of the stabilization performance achieved by calcium-based binders alone as stabilizers and calcium-based binders with calcined clay as a primary stabilizer.

## 2. Cement–Soil Stabilization

One of the most important and common techniques of chemical stabilization is mixing the soft soil with cement material to produce a soil–cement mixture, which contains soil, water, and measured amounts of cement and is compacted to the required density [31]. Many geotechnical problems are encountered when construction activities are carried out in soft soil deposits due to their high compressibility characteristics and low shear strength. Therefore, cement–soil stabilization has become a popular soft soil modification and stabilization technique in cement slurry or dry cement powder [32,33].

The modification of the soil–cement mixture occurs when Ca ions released from the cement during hydration and hydrolysis occupy the positions of exchangeable ions on the surface of the clay minerals, increasing stability and strength, controlling deformability, and reducing plasticity. At the same time, the stabilization of the soil–cement mixture occurs when cement is added to a reactive soil to generate long-term strength gain through cement reaction. This reaction generates stable cementitious products (calcium aluminate hydrates and calcium silicate hydrates) as the Ca from the cement reacts with the silicates and aluminates that are solubilized from the clay. As a result, cement treatment produces high and long-lasting strength gains [13]. Many researchers [9,12] have conducted experimental studies to identify how cement stabilization procedures can increase strength and compressibility in soft ground. Cement–soil stabilization has many benefits, such as decreased swelling and shrinking, increased strength and elastic modulus, and resistance to the damaging effects of moisture, freezing, and thawing. The cement additive decreased the maximum dry density and increased the optimum water content of sandy soils [34,35].Cement-treated soils are more brittle than untreated soils [36].

## 3. Lime–Soil Stabilization

Lime is made by heating limestone to very high temperatures. Three different forms of lime can be used to improve soil: hydrated lime (calcium hydroxide, Ca [OH]_2_), quicklime (calcium oxide, CaO), and hydrated lime slurry [37,38]. Lime is known to raise the soil’s shrinkage limit, optimum moisture content, and strength, while decreasing its liquid limit, plasticity index, swelling potential, and maximum dry density [4,5,19]. Lime also enhances the compatibility and workability of subgrade soils [39,40,41]. Soft soils benefit from lime stabilization in several ways, particularly in terms of its enhanced engineering properties, such as increased strength, less swelling, increased resilience, and resistance to the damaging effects of moisture. Clays with a range of plasticity, from medium to high, show the most improvement in these characteristics [42].

The optimum percentage of lime that increases the MDD, bearing capacity ratio, and strength, and decreases the plasticity indices is 5% lime by the dry mass of soil [43]. A total of 5% of lime is sufficient for cation exchange and pozzolanic reactions in soil and produces a new mineral (calcium aluminate hydrates) [44]. The optimum lime content for high strength has been shown to be 4–6% [45]. Lime-soil samples were prepared by many methods, such as injecting into the deep soil layers or mixing with soil in a dry state and adding water [46] (jet grouting, hydraulic, and deep soil mixing [47]), while lime columns were used for the shallow layers. Another method was lime slurry: A typical lime slurry for the stabilization of soil is made by mixing 1 kg of lime with 2.5 L of water, resulting in a 31% lime solution [48]. For example, according to the preceding guideline, 600 g of lime was combined with 1500 mL of tap water to make the lime slurry match the 6% dry soil weight previously compacted in the test mold.

## 4. Problems of Calcium-Based Binders

Environmental concerns pose a major threat to most countries worldwide, especially with the increasing infrastructure size. Cement is one of the most widely used construction materials. As a result, cement production and use have grown worldwide over time. In 1994, 1370 million tons (Mt) of cement was made worldwide, according to the United States Geological Survey (USGS) [49]. The USGS reports that global cement production has risen from 1370 Mt in 1994 to 4100 Mt in 2017, a more-than-threefold increase [50]. At the same time, the production of cement has received worldwide attention as one of the main sources of anthropogenic carbon dioxide emissions. The cement industry is a major cause of global warming [51]. It is considered the third largest industrial source of pollution, emitting more than 500,000 tons of nitrogen oxide, sulfur dioxide, and carbon monoxide, per year.

The associated CO_2_ emissions for clinker are between 849 and 868 kg CO_2_/ton. For OPC, the related CO_2_ emissions are between 802 and 855 kg CO_2_/ton [52]. The global process emissions in 2018 were 1.50 ± 0.12 Gt CO_2_, equivalent to approximately 4% of emissions from fossil fuels [22]. The cumulative carbon dioxide emissions from 1928 and 2018 were 38.3 ± 2.4 Gt CO_2_, 71% of which have occurred since 1990. Cement, hollow concrete blocks, and reinforcing bars (rebars) were the highest energy consumers and CO_2_ emitters in the study. They were accountable for 94% of the total embodied energy and 98% of the total CO_2_ emissions [53]. More information on CO_2_ emissions in specific regions was found by researchers of [54]. They studied the non-fossil fuel CO_2_ emissions from industrial processes in China between 2003–2018 for the production of lime, calcium carbide, plate glass, ethylene, aluminum, ferroalloys, soda ash, lead, and zinc. They showed that these industrial processes are equivalent to approximately 5% of China’s total CO_2_ emissions from cement production processes and fossil fuel combustion. In addition, a study on the CO_2_ emission factors for Chinese cement production based on organic and inorganic carbon from 2011 to 2015 [41] showed that the CO_2_ emission factor is 785.53–796.17 kgCO_2_/t_cl_.

Researchers [55] studied the CO_2_ emission factors of cement production in China and showed that the median values for the process, fuel, and direct emission factors are 525, 369, and 919 kg CO_2_/t clinker, respectively. However, the factor for electricity emissions is 74.9 kg CO_2_/t clinker. The final emission factor calculated from cement products is 761 kg CO_2_/t cement. Moreover, carbon dioxide emissions from 1574 cement factories in China were between 500 and 600 kg CO_2_/t clinker [56]. The carbon dioxide emissions from power plants varied among different enterprises, with an average level of 348 kg CO_2_/t clinker and a standard deviation of 233 kg CO_2_/t clinker. China’s cement companies, on average, emit 806 kg CO_2_/t clinker into the atmosphere. In 2009, the amounts of CO_2_ reached approximately 14.8% of the national CO_2_ emissions created by the cement industry in China [57]. Based on the current energy-related and emission-control policies, the researchers in [24] assessed the direct emissions of air pollutants from China’s cement industry, beginning in 1990, and forecasted future emissions through to 2020. The study showed that the cement industry produces approximately one-eighth of China’s total anthropogenic carbon dioxide emissions; emissions of CO_2_ increased 5.8 times in 2008.

Therefore, China is considered the largest producer and consumer of cement worldwide [23]. In 2010, China produced 1.87 billion metric tons of cement, about 57% of the total cement made worldwide. CO_2_ is released into the air in large amounts when fossil fuels are burned, and limestone is heated to create cement. In 2009, the cement industry released 1073 Mt of carbon dioxide into the air, representing 15% of China’s total greenhouse gas emissions. One study [58] compared different ways of calculating CO_2_ emissions from cement production to determine the uncertainties, finding that China’s cement-related CO_2_ emissions have a relative uncertainty of between 10% and 18%.

A study on the construction phase of a residential tower in Tehran Metropolitan City [59] found that the CO_2_ emissions were 6%, 78%, and 10% from cement mortar, concrete, and rebar, respectively. A study on the CO_2_ emissions in China [60] also showed that the CO_2_ emissions increased as cement production increased. Based on clinker output, raw material consumption (primarily limestone), fuel consumption (i.e., coal), and C/CR, the study displayed China’s cement CO_2_ emissions, by province, between 2005 and 2014. In 2005, cement production in China produced 641.31 Mt of CO_2_; in 2014, that number increased to 1246.04 Mt.

In Malaysia, the cement production is approximately 20 million tons per year [61]. The combustion of fossil fuels in pyro-handling units produces approximately 40% of the total emanations, while another 10% results from transporting crude materials and electricity. Finally, about 50% of carbon emissions are discharged in the decomposition of MgCO_3_ and CaCO_3_ to produce MgO and CaO and as the core chemical responses in the process. This study showed that, in 2006, Malaysia consumed 20 Mt of cement and had a clinker ratio of 0.89 t/t CO_2_, which is higher than the world average. Another study [25] analyzed the environmental impacts of the Brazilian cement industry, finding that fossil fuel production and clinkering together account for more than 70% of CO_2_ equivalent. Research on CO_2_ emissions in Poland’s cement industry [26] showed that the branch emissions index for Poland’s cement sector is 0.662 tons of CO_2_ per ton of produced cement. The concentrations of carbon dioxide emissions in Nigeria were recorded in the range of 2440–2600 mg/m^3^ [27].

The carbon dioxide (CO_2_) emission factor from lime applied in temperate upland soil was 0.026 mg C per mg of the CaCO_3_ emitted annually [62]. Furthermore, more than three billion metric tons of carbon dioxide are released into the earth’s atmosphere annually through cement, lime, and gypsum manufacturing enterprises [63]. The maximum global cumulative CO_2_ emissions related to the cement process would be 45.45 billion tons under the SSP3 scenario [64]. India, China, the United States (US), Nigeria, and Pakistan are responsible for the majority of the global total CO_2_ emissions from cement production processes between 2015 and 2000. In a new analysis of global process emissions in 2016 [18], the global CO_2_ emissions from cement production were demonstrated to be 1.45 0.20 Gt CO_2_, equivalent to about 4% of the emissions from fossil fuels. A total of 35% of the CO_2_ emissions come from fuel combustion to decompose and heat limestone to produce lime or clinker in an open atmosphere, and the remaining 65% comes from limestone rock itself [65]. An investigation of CO_2_ emissions [66] found that carbon emission levels in the cement industry range between 5% and 8%, with the plant producing approximately 900 kg of CO_2_ for every ton of OPC manufactured. This is similar to another study [67], where the manufacturing and use of the OPC used in concrete produced 810 kg CO_2_/ton of cement. Researchers [68] have shown that the carbon dioxide emissions and energy use from the worldwide cement industry were calculated to be 633 kg CO_2_/ton of cementitious product.

The researchers in [69,70,71,72] confirmed that 6–8% of the world’s ever-increasing anthropogenic CO_2_ emissions originate from the OPC industry. Approximately half of the 1435 Mt/ emissions are caused by the energy required to prepare the cement. The other half is unavoidable because it is a byproduct of converting CaO from CaCO_3_ and is intrinsic to binder chemistry.

Although lime is the second-largest source of CO_2_ emissions from industrial processes, after cement production in China’s lime industry, in general, about 800–850 kg of CO_2_ is released per ton of cement clinker. This represents about 5–8% of all CO_2_ emissions [73,74]. In total, the lime and cement industries were responsible for 8% of global carbon dioxide emissions between 2010 and 2011 [75]. Another study [76] showed that the process of emitting increased rapidly between 2001 and 2012, from 88.79 Mt to 141.72 Mt. The study’s emission factor and activity data have a relative uncertainty of 2.83 and 3.34 percent. Similar range of CO_2_ emissions from cement is given by [77]; cement plants account for approximately 5–7% of global CO_2_ emissions, with 900 kg of CO_2_ emitted into the atmosphere to produce one ton of cement. The cement industry produces about 5% of the global artificial carbon dioxide emissions, of which 40% is from burning fuel and 50% is from the chemical process [78]. The amount of carbon dioxide emitted by the cement industry is nearly 900 kg for every 1000 kg of cement produced. The high percentage of carbon dioxide produced in the chemical reaction leads to a large decrease in mass conversion from limestone to cement.

Carbon dioxide is emitted during the production process of non-metallic minerals, such as cement, plaster, lime, glass, and ceramics [79], which, in 2015, increased in the European Union (EU) by 2.5% compared with 2014. This is the result of widely varying trends in EU member states, with increases for the top seven emitters (with the exception of the decrease in Germany) being Germany (−0.2%), France (+2%), Italy, and Spain (+3.9%), Poland (+5.5%), Romania (+3.3%), and the United Kingdom (+3.1%). Carbon dioxide emissions are generated by the oxidation of carbonate in the cement clinker production process, which is considered the main constituent of cement and the highest of the non-combustion sources of carbon dioxide from industrial manufacturing, contributing to about 4.0% of the total global emissions in 2015. Fuel combustion related to cement production has a similar level for the emissions of CO_2_. Therefore, cement production accounts for about 8% of global CO_2_ emissions. Furthermore, in 2015, China produced 58% of the world’s cement, with India coming in second with 6.8% and the US coming in third with 2.7%. The EU is responsible for around 4.1% of the global CO_2_ output [80]. The carbon dioxide emissions for producing one ton of NaOH, lime, slag, and limestone used as raw materials in OPC were 2.987, 2.975, and 2.987 kg/CO_2_-e from limestone, slag, and various energies, respectively [81]. The emission factor for the carbon dioxide produced by cement manufacture is 0.82 kg CO_2_-e/kg [82].

Another major problem with calcium-based binders is the sulfate attack. When sulfates are in the soil or groundwater around a concrete structure, they seriously threaten its long-term durability. External sulfate can enter and cause a sulfate attack, one of the most well-known and studied chemical attacks. In OPC systems, sulfate attacks generally cause ettringite formation, accompanied by cracking, expansion, and a loss of strength. According to new theories, this expansion is driven by crystallization pressure when ettringite forms from an oversaturated solution in small pores. Near the material’s surface, gypsum and ettringite have been seen to form when a higher sulfate concentration is present. The temperature affected the durability of cement-based materials to sulfate attack, where the uniform surface crumbled at 5 °C and edges and corners scaled at 20 °C, and the damage was sharper at 20 °C than at 5 °C [83]. The researchers in [84] conducted a study to evaluate the effect of sodium and magnesium sulfate attacking OPC paste while an electric field was present. The results showed that ettringite was formed initially but broke down later to make gypsum. Thermodynamic modeling shows that the pore solution’s alkalinity dropped drastically during this process, causing ettringite to decompose. In addition to the sulfate access, decalcification occurred in this area, shown by the breakdown of portlandite and C-S-H. When the sample was exposed to MgSO_4_, the access to sulfate and decalcification occurred later and in a deeper area than when it was exposed to Na_2_SO_4_.

The results of thaumasite production by sulfosilicate clinker hydration [85] showed that the clinker’s belite and ternesite prefer turning into C-S-H gels, and the sulfate ions from the ternesite turn into gypsum. The chemical reaction between gypsum, carbon dioxide, and silica forms thaumasite. Thaumasite was clearly visible after 28 days of hydration, and its content in one sample reached approximately 34% in weight. The compressive strength first increased, and then decreased, within 56 days due to the sulfate attack [86,87]. When larger pores are filled with the products of erosion and develop into small pores in the early stage of erosion, in the later stage of erosion, the proportion of larger pores increases, and the cracks occur inside the specimen. Another study [88] showed that the acidic curing environment has a negative effect on the properties of concrete, where the strength decreased with an increase in the duration of the curing age and the proportion concentration of acid due to the sulfate attack.

Researchers [89] have investigated the degradation and mechanism of cast-in-situ concrete when immersed in sulfate-rich corrosive environments and found that corrosion in cast-in-situ concrete is much faster than the degradation of precast concrete due to the faster development of cracks in the cast-in-situ concrete. Sulfate attack leads to weight loss and great expansion in the later corrosion, putting concrete structures in great danger, particularly for cast-in-situ construction. The main products of corrosion induced by sulfate attack are gypsum and ettringite. A study [90] investigated the internal and external effects of sodium sulfate on the strength of soil–cement specimens and found that the internal attack decreased the strength of the soil–cement specimens by up to 70% compared with the prefabricated (about 40%) and cast-in-place (about 20%) samples.

This decrease in the trend of strength gain was seen after 28 days. The temperature and sulfate ion concentration in cement concrete sulfate attacks are highly significant, and the primary products of the erosion of sulfate attack on cement concrete are plate-like gypsum, rod-like ettringite with a larger slenderness ratio, incompletely corroded calcium hydroxide, and granular sulfate salt [91]. Under a seawater attack, the interaction between chloride ions and hydrates can form Kuzel’s and Friedel’s salts. Magnesium ion can replace the ion of Ca in portlandite, lowers the alkalinity of pore solution, and destabilizes the C–S–H gel. The change of phase primarily occurs on the surface of the concrete, weakening the structure and leading to delamination and spalling under the physical attack of the wave [92]. The production of gypsum and ettringite on the surface of the concrete causes a large tensile strength loss, which is the greatest threat to concrete structures in the field when subjected to a sulfate attack, particularly for OPC with a high C_3_A component [93]. Experimental studies on the paste of cement exposed to external sulfate attack (sodium sulfate solution Na_2_SO_4_) [94] found that the ettringite first precipitates in the largest pores without causing any expansion and then penetrates the gel and capillary pores, leading to accelerated swelling.

A further study [95] assessed the durability of a soil–cement mixture subjected to external sulfate attacks (a huge amount of sodium sulfates (25 g/L Na_2_SO_4_) was used to accelerate the degradation process). The results showed that for the most porous soil–cement specimens, an external sulfate attack could cause peeling on the surface of the test samples. Given the morphology of the needles seen by SEM analysis, this might be clarified through ettringite crystallization. On the one hand, the precipitation of these minerals results in a mass gain of up to 2.25% for some specimens after two years of sulfate exposure. The researchers in [96] studied the effect of environmental conditions on OPC structures and found that the cement with a higher amount of tricalcium aluminate (C_3_A) showed a more obvious deterioration. Visual changes, such as the crystallization of expansive products, cracking, and complete disintegration, were also observed. Furthermore, resistance was lost in specimens with low slag content. The loss of strength is a direct outcome of the sulfate attack because it causes a loss of cohesion due to the C-S-H decalcification. The cement with a lower percentage of CaO showed a better performance in resisting a sulfate attack. The generated stresses and the expansions in sulfate exposure conditions increased continuously with the increasing immersion time [97]. The effects of sulfate exposure on the pore network formation of different OPC matrices after two days of casting were investigated in [98]. The results indicate that the patterns of expansive product precipitation are related to the degree of refinement of the pore network. Large pores concentrate a greater proportion of the expanding product generated during the early phases of exposure. Later stages of precipitation result in finer pore sizes.

The production of gypsum and ettringite in the presence of sulfate ions had a beneficial effect on the evolution of the characteristics of OPC during the initial stage of a sodium sulfate attack [99]. Subsequently, these samples displayed a decline in characteristics due to the growth of expansive products and the development of microcracks. The mortar’s compressive strength and static elastic modulus grew during the initial immersion phase, and then plateaued as the immersion time increased. After an initial period of immersion (150 and 120 days), the static elastic modulus and compressive strength decreased as the immersion time increased. Similar behavior for strength was shown by another study [100]. The sodium sulfate solution-soaked mortar expanded and hardened to a higher density than the water-soaked type. The results also demonstrated that increasing the concentration of the sodium sulfate solution shortened the time required to achieve peak properties and accelerated the deterioration of the properties in the late stage.

A sulfate attack can generally be categorized into four phases [101]. In the first phase, ettringite forms at the expense of monosulfate (monosulfate dissolves continuously and is replaced by a greater volume of ettringite). The ettringite grows in stage two, at the expense of the carboaluminate phases (monocarbonate is destabilized and starts dissolving when the hemicarbonate is consumed, leading the ettringite growth to continue). Ettringite grows in stage three, at the expense of some hydrotalcite (hydrotalcite gradually dissolves in stages one and two, but in stage three it will be the only remaining solid that can contribute the aluminates to the solution). In the fourth phase, gypsum replaces portlandite. In addition, the calcium sulfoaluminate products form cement-stabilized clay as a result of the sulfate attack [102]. These products cause volumetric expansion and lead to the formation of microcracks in the specimens. Another study [103] found a sudden reduction in the small-strain shear modulus and a gradual increase in the hydraulic conductivity of a cement-admixed clay due to the degradation of the interparticle cementation in the specimens as a result of the sulfate attack. The damage is produced by ettringite production in pores as small as 10–50 nm, creating stresses of up to 8 MPa, exceeding the tensile strength of the binder matrix. Stresses increase with the C3A level and sulfate concentration [104]. In addition, the production of ettringite in micropores causes expansion due to the interaction of sulfate solutions with mortar, with the expansion reaching the crystallization pressure [97]. Crystals expand due to the pressure created by their growth within limited pores. The mortar bars exposed to a sulfate solution generated gypsum and ettringite, causing some expansion [105]. However, the bars eventually expanded and decomposed due to the development of thaumasite.

Cement clinker samples submerged in a sodium sulfate solution primarily revealed dicalcium silicate, tricalcium silicate, brownmillerite, and tricalcium aluminate as the primary mineral components. Ettringite is primarily formed when calcium aluminate reacts with sulfate ions. A sulfate attack, caused by ettringite expansion, manifests as microscopic fissures in the concrete. Sulfate ions can rapidly enter a solution at larger concentrations [106]. Gypsum forms in both high- and low-concentration solutions of MgSO_4_ and Na_2_SO_4_, and these compounds damage the C–S–H gel [107]. Under partial soaking conditions, cement mortar can be separated into four zones: the soaking zone, the wet zone, the crystallization zone, and the dry zone [108]. Corrosion products were studied in each zone. In the wet and soaking zones, ettringite is the most common by-product of corrosion. Gypsum and crystals of Na_2_SO_4_.10H_2_O and Na_2_SO_4_ are the corrosion products in the crystallization zone. Table 1 explains the major problems of calcium-based binders (cement and lime).

## 5. Alternatives and Partial Replacement by Calcined Clay

Cement in concrete is the most common artificial cementitious material worldwide, greatly impacting the world economy and environment. First, the energy consumption in producing cementitious material is high. An essential constituent of concrete releases a significant amount of CO_2_, a greenhouse gas. As illustrated above, 8% of the total carbon emissions come from cement material. Therefore, enhancing the sustainability of cement concrete has become a significant issue in recent decades to improve sustainable development. Globally, improving cement concrete’s sustainable characteristics and low-carbon content has been the focus of industry attention in recent decades.

Many methods are available to enhance cement concrete’s sustainability and reduce cement material’s impact on the economy or environment, such as using chemical and mineral admixtures in the concrete. One of these methods is the use of calcined clay (natural pozzolana) in concrete, which has developed rapidly in recent years, as demonstrated by the researchers reviewed in Table 2. The table summarizes the most recent research on using calcined clay as a partial replacement for cement or lime in cement and soil improvement. Figure 1 shows the effectiveness of calcined clay on compressive strength as an alternative for lime or cement in concrete. This calcined clay material can be found both artificially and naturally.

Using calcined clay (CC) offers significant advantages as a cement replacement material and a low-cost alternative binder. Using CC could enhance the strength, which reached an approximate 10% increase in compressive strength relative to the control at 90 days [168]. Its use also reduces water adsorption [166], reduces the coefficients of chloride ions [161], and increases the durability of concrete [134,140]. Importantly, using CC is considered a good solution to reduce carbon dioxide emissions and produce an eco-friendly CC at a low cost [151]. The replacement proportion in cement clinker can reach as low as 50% [132,138]. Limestone–CC cement has exceptional durability and mechanical properties, such as a higher final compressive strength. In addition, it increases the C-S-H amount at 28 days of hydration [167]. 

As a result, CC can be considered a global alternative to a variety of traditional low-carbon OPC materials. This “pozzolanic calcined clay,” as it is also known, in addition to reducing CO2 emissions in industries, also brings several economic advantages for cement production. Its application in South Asia and Latin America is gradually growing.

## 6. Conclusions

This paper has focused on two main parts. The first deals with the problems of using cement and lime as supplementary cementitious materials (SCMs) cause for soil, concrete, and the environment. For example, where the energy consumption involved in producing cementitious materials is high, carbon dioxide emissions account for 8% of the total carbon emissions. The presence of sulfates in the soils or groundwater surrounding a concrete structure may also cause a serious threat to the long-term durability of concrete and soil due to the effect of a sulfate attack, which is considered one of the most widely admitted and well-studied chemical attacks.

Therefore, improving the sustainability of cement concrete has become a major issue in improving socially sustainable development. Internationally, in recent decades, improving cement concrete’s sustainable and low-carbon characteristics has become the focus of industry attention by using SCMs, which are industrial by-products or natural materials used to improve the durability, performance, and sustainability of concrete mixtures. One of these SCMs is calcined clay; therefore, the second part of this review paper focuses on using this material to reduce the use of cement or lime by comparing cement or lime alone and using calcined clay with cement or lime.

Based on the results of using calcined clay in concrete, which has developed rapidly in recent years, this material, can produces a low-carbon cement-based material, and it is recommended for use in concrete. Compared with traditional OPC, the clinker content in its cement can be as low as 50% because it uses a large amount of calcined clay and limestone as key components. It will conserve the limestone resources used in cement manufacturing and help to reduce the carbon footprint associated with the cement industry. Its application in places such as Latin America and South Asia is gradually growing.

## Figures and Tables

**Figure 1 materials-16-02020-f001:**
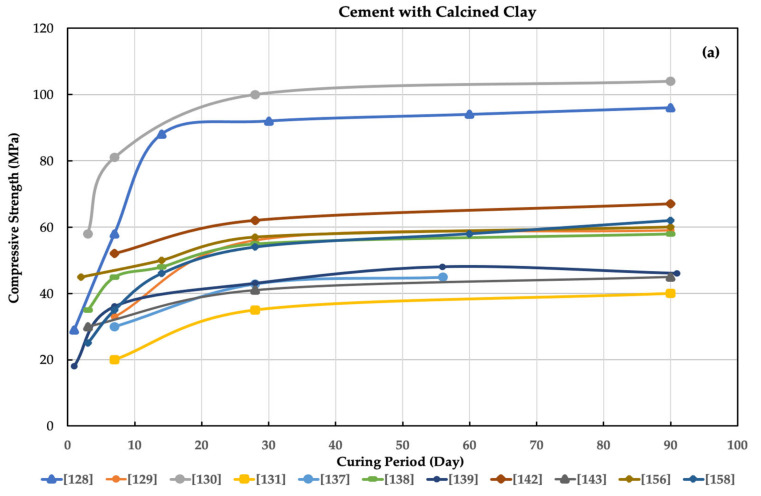
Compressive strength for (**a**) cement only and (**b**) cement with calcined clay as alternatives for cement in concrete by many researchers.

**Table 1 materials-16-02020-t001:** Calcium-based Binders (Cement and Lime) Problems.

Type of Calcium Binder [Ref.] (Problem)	Findings
Lime [62] CO_2_ emission	The CO_2_ emission was 0.12 Mg per Mg for CaCO_3,_ which indicates that 100% C is ultimately released into the atmosphere in the form of CO_2_
Lime [109] Sulphate Attack	The ettringite formed and caused swelling with a high affinity to absorb water, causing a decrease in compressive strength and destroying the structure, especially in earlier stages of formation.
Cement [110] CO_2_ emission	A considerable share of global CO_2_ emissions comes from OPC production.
Cement [18] CO_2_ emission	- Cement production is the third most significant source of anthropogenic CO_2_ emissions. Cumulative emissions were 39.3 ± 2.4 Gt CO_2_ from 1928 to 2016, 66% since 1990.
Cement [111] Sulfate attack	- Due to the high solubility of gypsum in water. The great molar volume of ettringite reinforces internal stress in the cementing matrix, and this cause an expansion. The more SO_3_ is added, the more time is given for the formation of ettringite, where for 2% added, a large amount of ettringite is formed.
Lime [112] CO_2_ emission	- The production of lime is the second highest source of carbon emission from industrial processes. The emission of Carbone dioxide increased speedily from 88.79 million tons to 141.72 million tons from 2001 to 2016 in China’s lime industry.
Cement [113] CO_2_ emission	8% of anthropogenic CO_2_ emissions are generated in the global cement.
Lime [114] Sulfate attack	The sulfate caused the swell potentials and plasticity to increase unusually because of the formation of the ettringite minerals. In addition, the shear strength decreased with increased sulfate concentration and curing time.
Lime [115] Sulfate attack	The sulfate in the soil can react with the hydraulic binder and the aluminosilicates to form expansive minerals.
Cement [116] CO_2_ emission	Around 6% of all artificial carbon emissions are produced by every ton of OPC.
Lime [117] Sulfate attack	Samples containing sulfate and lime experienced swelling due to the ettringite formation in the samples. Any presence of sulfate in the natural soil could produce ettringite if calcium-based stabilizers are used.
Lime [118] Sulfate attack	The ettringite formation in the sulfate clay system negatively affects marine clay engineering properties.
Lime [119] Sulfate attack	In the presence of sulfate, the shear strength initially increases with a cure period, then drastically decreases after cure after more than 180 days due to ettringite formation.
Cement [17] CO_2_ emission	The emission of CO_2_ in the cement industry is from two parts: raw and fuel burning; CO_2_ emissions represent approximately 5–7 % of global emissions of CO_2_.
Lime [120] Sulfate attack	Sulfate levels cause abnormal changes in the volume of lime-stabilized soil and reduce the shear strength of lime-treated black cotton soil after long treatment periods. However, the effect of sulfate is marginal for short healing periods.
Lime [121] (Sulfate attack)	the effects of sulfate depend on the type of sulfate cation. Ca^2+^ and Mg^2+^ increase the lime-added effect on the consistency and dynamic compaction properties of clay. Others tend to reverse these effects, Na^+^ and K^+^.
Lime [122] (Sulfate attack)	Results showed that the higher gypsum levels (up to 8 WT) resulted in significant water absorption, extreme expansion, and high inflationary pressure due to ettringite formation.
Lime [123] (Sulfate attack)	Whenever there was a sulfate, ettringite formation was present in all lime-treated samples.
Lime [124] (Sulfate attack)	After several years in a specific case study, lime-treated sulfate-bearing clay swelled and disintegrated when used for road building. Abundant thaumasite, a complex mineral of calcium-silicate-hydrates, is found in heavy areas.
Cement [125] (Sulfate attack)	The results show no direct correlation between the degree of expansion of cement on sulfate attack and the amount of crystalline calcium sulphoaluminate present. Other factors, such as its stability under prevalent conditions and the influence of other ions, particularly magnesium and chloride ions, may predominate. In addition, protective surface films also play a significant part.
Lime and cement [126] (Sulfate attack)	In samples of 10% lime-treated heavy clay and at constant moisture content for 1 week, swelling and cracking were observed when immersed in magnesium sulfate or sodium sulfate solutions at levels less than 1.5% as SO_3_.

**Table 2 materials-16-02020-t002:** Calcined Clay as a Partial Replacement for Cement or Lime in Cement and Soil Improvement.

Optim. Mixture [Ref.]	Major Properties Tested	Cement only	Cement with CC	Findings	Future Perspectives
OPC + 30% CC [127]	Hydration degree (CH[g/gC3S reacted])	0.45	0.3	The aluminate and silicate clinker reactions are affected and accelerated by the SCMs, but in varying ways and to varying degrees, such as enhanced initial ettringite formation and initial dissolution of C3A.	The partial replacement of cement with CC has the greatest promise as a worldwide short-term solution to substantially reduce cement producers’ greenhouse gas emissions.
	Ettringite (Wt. %)	9.2	6.8		
Cement + 33% CC +16.67% LS [128]	Compressive strength MPa at 28 d.	68	55	Although the compressive strength is not visibly enhanced by adding calcined clay and limestone powder as a 50–70% substitution for cement, these additives considerably increase the toughness, densify the microstructure, and refine the pore structure of cementitious materials.	Reduced clinker use may benefit the cement industry both environmentally and economically. In underdeveloped nations, cheaper cement mixes will help infrastructure development and reduce greenhouse gas emissions.
	Flexural strength MPa at 28 d.	9.4	9.7		
Cement + 30% CC [129] Cement+ 10% LS + 20% CC [129]	Slump (mm)	140	100–110	The compressive strength of binary and ternary blends was between 9.0–14.3% and 18.2–28.7% lower than that of the OPC mixture. The permeability of binary and ternary blends containing CC (with and without using LS powder) was lower than the control mixture, and 30 wt% replacement of Portland cement with CC and LS powder caused to decrease in the Dapp and Dnssm of binary and LC3 concretes.	-
	Measured unit weight of concrete (kg/m)	2313	2291		
	Compressive strength MPa at 360 d.	64	60		
	Water absorption (wt.%) at 28 d.	3.3	3.1		
	Water penetration depth mm at 28 d.	8.3	4.5		
	Surface electrical resistivity (k Ω-cm)	10	35		
	Non-steady-state migration diffusion coefficient (Dnssm)	18	10.6–6		
	Slump (mm)	140	105–120		
	Measured unit weight of concrete (kg/m)	2313	2318		
	Compressive strength MPa at 360 days	64	55		
	Water absorption (wt.%) at 28 days	3.3	3.05		
	Water penetration depth mm at 28 days	8.3	5		
	Surface electrical resistivity (k Ω-cm)	10	30		
	Non-steady-state migration diffusion coefficient (Dnssm)	18	14–8		
OPC + 22% LS + 45% CC (2:1) ratio [130]	Compressive Strength mPa at 28 days	95	100	A large number of amorphous C–S–H, CH, and thin plates can be observed by FESEM image for the mixture at 28 days. The compressive strength increased significantly, with the addition of CC and the 33 % BRC specimens showing higher strength than the 0 %BRC specimen.	It reduces CO_2_ (greenhouse gas) emissions and promotes sustainable development.
	Ultrasonic pulse velocity (m/s)	4625	4683		
Cement + 30% LS+ 30% CC [131]	The packing density of mortar ΦM	0.815	0.816	It has been possible to prepare ternary CEM I/ CC/L binders for mortars featuring an adjusted spread and a compress- save strength close to 32.5 MPa at 28 days.	The partial replacement of cement with a combination of CC and LS fillers is a promising method for reducing the environmental effect of concrete, enhancing its long-term mechanical performance and durability.
	Compressive strength of mortars at 28 days mPa	53	32.5		
Cement + LS + 30% CC [132]	Particle density [g/cm^3^]	3.07	2.94	The using of CC causes a significant increase in yield stress, viscosity, and four times flow resistance compared to PLC.	
	Total surface area [m^2^/cm^3^]	5.5	10.1		
	Water demand [wt%]	26.6	29.1		
	Viscosity factor [Nmm*min]	0.11 ± 0.01	0.16		
	Yield stress factor [Nmm]	12.2 ± 0.7	66.9		
	Flow resistance [Nmm/min]	1987 ± 69	7841		
OPC + 30%CC + 15% LS filler [133]	Bound water (g/100 g _anh, binder_)	25	23	Significant pozzolanic activity and synergy between LS filler and CC were seen in composite CC containing a minor amount of metakaolinite. In a ternary system, the products and degree of hydration are expected to be the same for CC with a low metakaolin content as for CC with a high metakaolin content.	It has been verified that the combination of Portland cement, calcined clay, and lime- stone filler is a promising way to maximize the potential usage of composite clays in cement-based composites.
	Portlandite content (g/100 g _anh, binder_)	14	9		
	Degree of hydration	0.55	0.88		
Soil + 6 % Cement or lime Soil + 6% CC [134]	Liquid limit (%)	59–57	54	The application of calcined clay led to a better compaction property. The increase in CC led to an increase in the specific gravity and maximum dry density in each mixture. At the early stages, the pozzolanic reaction was dominated considerably by the hydration of calcium hydroxide with alkali-exchanged clinoptilolite, carbolaluminate hydrate groups were the preliminary product of this hydration.	Adding zeolite and lime to fine sand engineering is a unique method for changing the grain size distribution of poorly graded soils by adding fine filler content. At the same time, zeolite, as a natural pozzolan in combination with calcium hydroxide, may also induce artificial cementation.
	Plastic limit (%)	43–38	19		
	Plasticity index	16–19	35		
	Swell percent	4.57–0	6.1		
	Swell pressure (kPa)	116–0	161		
Soil + 3% lime Soil + 3% natural pozzolana [135]	Shrinkage (%)	13	9	The 4% proportion reacted, but 3% of natural pozzolan alone showed no sign.	-
	PI (%)	36	27		
	OMC (%)	34	34		
	MDD (kN/m^3^)	21.3	21.3		
	Stress (MPa)	1.05	1.04		
OPC + 15% LS. powder or + 30% CC [136]	Compressive strength (MPa)	49	65	At 300 °C the strengths of all samples increase, while those of the LC3 ternary blended pastes increase significantly more because of further hydration of binders and the formation of katoite	-
OPC + 8% calcined Phyllite rock [137]	Compressive strength (MPa)	33.8	42.7	Higher resistance to chloride ion penetration when adding CC to concrete. The durability of 8% Calcined phyllite concrete is better when compared to the reference mix.	-
	Flexural strength (MPa)	4.2	5.1		
	Rapid Chloride Ion Penetration (coulombs)	2411	453		
	Slump (mm)	135			
OPC + (50–60) % LS + CC (LC2) cement [138]	Compressive strength (MPa)	62	59	Cement with 50%, 60%, and 70% Limestone-calcined clay gives a compressive strength of 53.6, 43.9, and 33.4 MPa after 28 d., respectively; thus, they fulfill the requirements of 28-day strength for 52.5, 42.5, and 32.5 N cement, respectively.	Cement with LS-CC (50, 60%) shows lower embodied energy and carbon emission indices. These results can help the construction industry reduce its carbon footprint.
	Embodied energy (MJ/kg cement)	5.5	4.2		
	Carbone emission (kg CO_2_/kg cement)	0.92	0.56		
OPC replaced by 30% of CC + LS in a 2:1 wt ratio [139]	Portlandite (%) after 28 days	16.53	11.6	Strength when using CC with OPC is closer to control OPC	CC can be used as a viable alternative to replacing cement and produce a low-carbon and sustainable concrete
	Bound water (%)	14	11		
	Compressive strength (MPa)	54	46		
	Rapid chloride penetration (coulombs)	4700	6500		
PC + 15% CC [140]	Portlandite contents (CH), %	11.4	8.6	The latter enhancement in the cement mortars blended with CC was due to the refinement of the pore structure, compared to the Portland cement mortar. The mixed cement mortars had lower carbonation resistance than the ordinary Portland cement mortar.	CO_2_ emissions from cement and concrete production can be reduced by replacing some Portland cement with these SCMs.
	Slump flow (mm)	184.5	161.5		
	Compressive strength (MPa)	68	55		
	Average carbonation depth, *d*k (mm) (after 270 day exposure)	3.5	7		
OPC + 15% LS or 30% CC [141]	Compressive strength (MPa)	53	45	For all mixtures aged 3–270 days, the combined water strength increased linearly with time. As LS and CC replacement levels were raised, there was a corresponding rise in electrical resistivity.	Hwangtoh calcined clay is a type of kaolin clay that is used in construction as an eco-friendly material. In contrast to other SCMs, it can be used as an environment-purification material.
	Bound water per gr binder	26.5	23		
	Electrical resistivity (kΩ.cm)	75	180		
OPC + 50% clinker + 30% CC [142]	Compressive strength (MPa)	62	62	The results explained the impact of CC on increased superplasticizer demand and show the difficulties in retaining the workability for extended durations.	-
	Viscosity (Pa s)	22	55		
OPC + 15%LS +31% CC replacing the OPC [143]	Slump (mm)	90	120	LC3 concrete systems had an order of magnitude resistivity greater than OPC concrete at both early and later ages. Compared to fly ash concretes, concretes made with CC reach the critical pore size and densify the capillary pore space early.	-
	Surface resistivity (kohm.cm)	15	270		
	Compressive strength (MPa)	55	45		
	Conductivity (S/m)	0.04	0.001		
	Pore solution conductivity (S/m)	5.17	1.43		
	Tortuosity	9.65	27.93		
	Porosity %	7.6	8.3		
OPC + 10% LS + 10% CC [144]	Compressive strength (MPa)	89	94	The corrosion value on the surface of carbon steel was lower in concrete specimens containing CC and LS admixtures. The value of double-layer capacitance was reduced for the concrete with CC and LS, the passive layer thickness was enhanced and, resulting in an improved protective capacity.	-
	Water absorption (%)	8.8	4.8		
	Corrosion rate (MMPY)×10^−3^	2.8	0.82		
OPC + 5% CC [145]	Density (kg/m^3^)	2.37	2.43	No significant effect on the workability of mortar and higher strength was achieved at OPC replacement with 5% CC content.	Calcined clay was suitable for improving the properties of lightweight mortars.
	Compressive strength (N/mm^2^)	32	23		
Cement-LS with 30% CC cured in sulfate [146]	Compressive strength (MPa)	9	45	The CC pozzolanic reaction in cement mortars is similarly developed in aggressive and non-aggressive curing conditions, where pore size refinement, consumption of CH, and prevention of the sulfate ingress. Cement with 30% CC greatly resisted ESA, while a worse performance was presented for limestone cement.	-
	expansion (%)	0.55	0.005		
OPC + 30% CC [147]	Chemical shrinkage, mL/g cement	0.08	0.12	Calcined clay may start the Pozzolanic reaction within the first day. The mechanical properties of concrete with the blended binder are lower than those of pure cement mix in the first three days but increase faster and become comparable from 7 days onwards.	Comparable mechanical properties and higher endurance indicate that it is possible to produce high-performance concrete with a significant proportion of clay and limestone.
	Cumulated heat, J/g cement	330	400		
	CH content, %	20	12		
	Compressive strength (MPa)	89	92		
	Elastic modulus, GPa	41	43		
	Drying shrinkage,10^−6^	370	240		
OPC + 20%CC [148]	Alkali-Silica reaction (ASR) with NaOH	0.33%	0.12%	A pavement-grade concrete mixture (cement replacement by 20%CC by weight) gave fresh and hardened air content and desired workability. The strength development of this mixture was slightly below the control. The use of CC improved the durability of concrete concerning alkali–silica reaction, chloride penetration, and drying shrinkage compared with the control mixture.	Limestone-calcined clay–cement and slag -calcined clay–cement mortar mixes exhibited great strength development after substituting about 50 percent of the Portland cement. SCMs are essential components of modern concrete and are used to increase workability and durability (e.g., embodied energy and CO_2_ reduction).
	Compressive strength (MPa)	27	24		
	Slump (cm)	11.4	12.4		
	Fresh density (kg/m_3_)	2244	2214		
	Fresh air content (vol %)	6.5	6.8		
	Hardened air Content (vol %)	6.3	7.1		
	Air spacing factor (mm)	0.151	0.144		
	Drying shrinkage strain %	−0.09%	−0.092%		
Cement slurry + (2) % CC [149]	Shear stress (Ib/100 ft^2^)	80	118	It enhanced the rheological properties of cement slurries. The plastic viscosity rose with a rise in percentages of CC. Higher values of yield point. CC- based cement slurries resulted in robust structure and sustained high compressive strength after exposure to high-temperature, high-pressure conditions. The compositional analysis showed that CC gave a strongly bonded structure and a low calcium/silica ratio.	This work will pave the way for future research on using CC in oil and gas cementing and its durability over time.
	Plastic viscosity (cP)	64.4	94.7		
	Yield point (Ib/100 ft^2^)	24.9	35.1		
	Un API Fluid Loss (mL)	2091	1980		
	Uniaxial Compressive Strength (UCS)(psi)	4776.5	5895.2		
OPC + 50 wt.% [150]	Specific surface (cm^2^/g) (Blaine)	3210	3990	The mass loss and the autogenous-drying shrinkage of concretes containing cement blended with CC were lower than concretes containing PC. The concrete compressive strength was improved up to a 50% replacement ratio.	-
	Density (g/cm^3^)	3.12	2.76		
	Volume expansion (mm)	2	5.5		
	water demand (%)	0.32	0.4		
	Compressive strength (MPa)	52	57		
	ultrasonic pulse velocity (m/s)	4440	4620		
	energy demand (kWh/t)	1000	770		
	mass loss (%)	3.03	2.08		
	dry shrinkage (×10^−6^)	610	500		
OPC + 15 % calcined bentonite (CB) [151]	T500 flow time (sec)	1.9	3.4	The addition of calcined bentonite reduced the fresh properties, slump flow, and flow times of SCC. The results of segregation testing are good for SCC manufacture. SCC with 10–15% CB exhibited a greater compressive strength up to 90 days after hardening. The addition of calcined bentonite improved gas permeability, porosity properties, and chloride-ions penetration.	CB is a good solution that will reduce CO_2_ emissions and produce eco-friendly at a low cost and durable SCC.
	Slump flow diameter (mm)	710	750		
	Segregation index (%)	11	8		
	Compressive strength (MPa)	62	74		
	Apparent gas permeability Kapp (*10−6 m2)	0.38	0.25		
OPC + 15% LS-CC (LC3) [152]	Compressive creep compliance [µm/m/MPa]	118	100	Lower creep. The elastic characteristics of C-S-H were found to be comparable in ordinary cement and LC3. The viscosity behavior of C-S-H gel appears to be considerably different for LC3. The greater viscosity of C-S-H gel in LC3 might be attributable to a chemical composition difference.	-
	C-S-H gel (GPa)	23	26.7		
Portland cement + Calcined Shale CS [153]	Compressive strength (CS) (MPa)	55	52.4	Calcined illite-chlorite I/Ch shale was good strength at 90 days.	The use of CS reduces CO_2_ emissions in cement and concrete industries.
	Strength activity index (SAI)	1	1		
	Flow, %	142	134		
OPC + 30%CC + 15% LS. [154]	Degree of hydration of belite	0.82	0.38	Using CC helped achieve a well-refined microstructure in LS-CC cement within 7 days of hydration. The presence of CC caused a decrease in the degree of hydration of clinker phases.	-
	Degree of hydration ofalite	0.96	0.84		
	Compressive strength (MPa)	50	45		
OPC + 31% CC, 15% LS [155]	Compressive strength (MPa)	60	56.7	The carbon footprint of limestone calcined clay cement (LC3) concrete was much lower than that of OPC concrete of comparable strength.	The using of CC has shown to be a good solution to reducing CO_2_
	Diffusion coefficient (×10^−12^ m^2^/s)	15.6	1.7		
	Electrical conductivity mS/m	6.23	0.14		
	Ageing coefficient, m	0.17	0.54		
	Total CO_2_ emissions/m of concrete (kgCO_2_ eq./m)	380	270		
OPC + 30% CC [156]	Compressive strength [N/mm^2^]	62	62	At this proportion, the concrete properties were not changed to a significant extent. The cement with CC performed better than the reference in inhibiting durability issues, alkali-silica reaction (ASR), chloride migration, and sulfate resistance. However, negative effects were found on the carbonation velocity and the early strengths. For the majority of concrete applications.	-
	Carbonation depth [mm]	4.5	10		
	Expansion [mm/m]	2.2	0.2		
	Chloride migration coefficients DCl- [10−12 m^2^/s]	8.2	2.7		
OPC + 15% LS + 30% CC LC3 [157]	Specific gravity	3.15	3.12	LC3 and OPC develop comparable strength, higher strength sows in, and LC3 contains superior quality of limestone to that of OPC. The optimum dosage is 15% limestone, 30% calcined clay, and 5% gypsum; a higher substitution of 35% also shows comparable strength development.	CC is now a common SCMs used to reduce cement use (to replace clinker up to 40–50%).
	Standard consistency	31%	37%		
	Initial setting time	44	98		
	Final setting time	348	410		
	Compressive strength (MPa)	41	40		
Cement +20% CC [158] 25%, CC	Compressive strength (N/mm^2^)	21.5	28	The material has shown the potential to mitigate carbon emissions by replacing cement by as much as 20 to 50%.	CC is a suitable additive for reducing carbon emissions without compromising strength improvement.
	Compressive strength UCS (N/mm^2^)	2.94	4.64		
OPC + 20% CC +15%LS+ 5% gypsum [159]	Compressive strength (MPa)	62	57	It shows approximately equal strength at 28 days compared to ordinary Portland cement. Even after curing with 2% of acid and sulfate dilutions, good compressive strength is shown when using the additives. Durability was increased as CC content increased,	The combination of 50% clinker, 15% LS, 30% CC, and 5% gypsum is a modern cement. Here, clinker is decreased by 50%, resulting in a 30% reduction in CO_2_ emissions.
OPC + 40% of 2:1 (CC to LS) [160]	Flexural strength (MPa)	8	9.7	The flexural strength increased significantly due to the greater formation of crystalline aluminates in the LC3.	-
	Compressive strength (MPa)	45	40		
OPC + 20% CC [161]	SO_3_	0.52	0.46	Using CC improves the compressive strength, fresh condition, durability, and microscopic structure of ordinary concretes. Decrease in capillary water absorption and increase in compressive strength from 28 days to 90 days. The coefficients of chloride ions in concrete with CC are low when compared with concrete without CC. Reductions in the chloride penetration depth were also observed.	-
	CaCO_3_	47	40		
	Oven dry density (g/cm_3_)	2.37	2.34		
	Water porosity (%)	13.3	11.38		
	Compressive strength (MPa)	37.23	32.39		
	Pozzolanic activity index	1	0.78		
	Absorption (g/mm2)	2.1	1.7		
	Chloride ion concentration (mol/1)	0.006	0.004		
	Mass loss (%)	−0.7	−2.7		
OPC + 21% CC–LS –exposed to a 0.11 M Na_2_SO_4_ [162]	Compressive Strength (MPa)	69	72	The findings indicate that all mortars with CC/(CC + L) 0.5 have high sulfate resistance.	It can suggest from the results that the Portland cement–CC-Limestone is included as a new form of sulfate-resistant Portland composite cement and Portland pozzolana cement by industry standards.
OPC + 45% CC [163]	Final Setting time (mint.)	180	240	The chemically activated cement shows lower porosity, higher pozzolanic activity, higher resistance to acid attack, and shorter setting times compared to non-activated cement.	-
	Compressive Strength MPa at 90 days	48	44		
	Porosity %	22	15		
Cement + 15%LS + 30% CC [164]	Porosity %	26%	24%	Higher substitution levels are possible with a combination of CC and LS to around 50% with similar mechanical properties and durability.	CC results in a smaller carbon footprint and lower environmental impact.
	CO_2_ emissions for concretes of 30 MPa grade kg CO_2_eq./kg	0.145	0.105		
	CO_2_ emissions for concrete of 50 MPa grade kg CO_2_eq./kg	0.175	0.11		
OPC + 15%LS +30% CC [165]	Specific gravity	3.16	3.01	This cement paste research has proven that the LC3 cementitious system can produce more durable concrete than either OPC or the widely-used fly ash-based PPC.	The key to improving the environmental friendliness of cement is using mixes such as these, which have a low clinker content but significant performance implications.
	Consistency (%)	30	33		
	Initial setting time (min)	124	101		
	Final setting time (min)	245	165		
	Blaine’s fineness (m^2^/kg)	340	520		
	Compressive strength of cement at 28 days (MPa)	61	42.1		
	Intrinsic permeability of hydrated cement paste at 28 days (10^−20^ m^2^)	2	0.04		
Cement + 15% CC [166]	Compressive strength MPa	57.3	77	It has been found that adding calcined marl to Portland cement increases its compressive strength (from 5% to 37%), density, and water resistance (from 0.92 to 0.93–0.98). In addition, once calcined marl was included, the water adsorption values dropped from 1.0 to 0.9–0.7.	The Portland cement pastes enriched with the addition of 10–15% marl calcined showed the best properties.
	Normal consistency	27.3	30.4		
	Density, kg/m^3^ /%	2270	2300/1.3		
	Water adsorption, %	1	0.90		
	Water resistance	0.920	0.980		
	Specific surface area, m^2^/kg		800		
OPC + 3% CC [167]	CH content/%	23.9	19	The loss of mass and the reduction in compressive strength were significantly less for specimens with the addition of CC than for the control mortar specimens. Using CC reduced the CH content while increasing the C-S-H amount at 28 days of hydration.	-
	Intensity/counts at 28 days	1290	900		
	C-S-H/%	65.8	76.7		
	Unreacted/%	10.005	6.181		
	Porosity/%	0.17	0.003		
	Compressive strength loss, Dfc% at 60 days	52	40		
Cement + 15 %CC [168]	Compressive Strengths at 1,	18	20	The CC consumed higher portlandite, and the compressive strength increased when the amorphous content of the CC increased. The CC reached an approximate 10% increase in compressive strength relative to the control at 90 days.	Using CC offers significant advantages as a cement replacement material, a low-cost alternative binder, with the ability to enhance strength.
	3,	28	30		
	7,	34	38		
	28	38	36		
	and 90 days of curing	42	46		
Portland Cement + 30%CC [169]	Compressive strength MPa			The Pozzolanic reactivity is related to the specific surface area of CC. The specific surface area of CC is related to the calcination degree. The addition of CC produces the acceleration of cement hydration, improves the volume and structure of the pore system in the paste	Using the suitable blended cement mix with CC makes it possible to reduce carbon dioxide emissions and improve mechanical and durability performance.
	at 2 days	25	23		
	At 7 days	30	35		
	At 28 days	37	48		
	[Cao] mmmol/l				
	at 2 days	8	2		
	At 7 days	6	1		
	At 28 days	5	0.5		
	[OH] mmol/l				
	at 2 days	85	48		
	At 7 days	96	45		
	At 28 days	106	50		
	Sorptivity Coefficient	0.094	0.022		

## Data Availability

All the data reported in this study was originally generated and can be requested from the corresponding author.

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
