# Peer review of "Calcium-Based Binders in Concrete or Soil Stabilization: Challenges, Problems, and Calcined Clay as Partial Replacement to Produce Low-Carbon Cement"

_materials, 2023, doi:10.3390/ma16052020_

Round 1

Reviewer 1 Report

Journal: Materials – ISSN 1996-1944

Manuscript ID: Materials-2142742

Title: Review of Calcium-Based Binders in Soil Stabilization- Challenges – Problems-Calcined Clay as Partial Replacement to Produce Low-Carbon Cement

I have carefully reviewed the article and esteemed to say that authors have covered a new review type which covers a lot of topics to which I suggest it to get published the way it is with minor corrections and changes as mentioned below.

1.     Authors have to carefully look for grammatical mistakes as the whole article is covered with major English corrections it is requested to alter using an expert.

2.     In introduction authors have to cover with a deep briefing in the emission on CO2 and its effectiveness towards different countries with a statistical point referred by the WHO

3.     In lime-soil stabilization authors must add a few more lines regarding the plasticity index, dry density and how the slurry is being prepared kind off.

4.     It is requested to add a table mentioning different calcium-based binders and their major problems with effective mass approximation concept.

5.     A pictorial representation would be good to understand more about the alternatives of the calcined clay

6.     Authors can add a new column stating the future perspectives and its importance in different fields.

7.     Authors can site the below papers stating in introduction.

8.     Darange, R., Adesina, A., & Das, S. (2022). Feasibility study on the sustainable utilization of uncalcined clay soils as Low-Cost binders. Construction and Building Materials, 340, 127724.

9.     Sharma, T. S. K., & Hwa, K. Y. (2022). Architecting hierarchal Zn3V2O8/P-rGO nanostructure: Electrochemical determination of anti-viral drug azithromycin in biological samples using SPCE. Chemical Engineering Journal, 439, 135591.

Gautam, S., Kumar, A., Vashistha, V. K., & Das, D. K. (2020). Phyto-assisted synthesis and characterization of V2O5 nanomaterial and their electrochemical and antimicrobial investigations. Nano Life, 10(03), 2050003.

Reviewer 2 Report

Respected Authors

It may be easily noticed that your study was originally formatted to another template. The way you address references and the format of the reference list should be adjusted to MDPI rules. Please avoid "cluster citations" as every cited paper deserves some words of introduction to prove their relevance and importance for the current study.

Your work looks slightly chaotic in my eyes. You focus on soil stabilization but most of presented results refer to concrete with high compressive strength. Most of presented results are obtained after a relatively short time (up to 28 days). From my experience with backfill materials in mining industry, a well-defined research program concerning soils and/or tailings stabilized with hydraulic binders should last non less than 2-3 months. Just to check the development of strength and stiffness in time and notice possible degradation caused by aggressive factors (sulphate). I made some reference suggestions below. None of the papers is mine nor come from my affiliation so there is no personal gain behind it. Please consider:

1. Egorova, A.; Rybak, J.; Stefaniuk, D.; ZajÄ…czkowski, P. Basic Aspects of Deep Soil Mixing Technology Control. IOP Conf. Ser. Mater. Sci. Eng. 2017, 245(2), 022019. DOI 10.1088/1757-899X/245/2/022019

2. Kanty, P.; Rybak, J.; Stefaniuk, D. Some Remarks on Practical Aspects of Laboratory Testing of Deep Soil Mixing Composites Achieved in Organic Soils. IOP Conf. Ser.: Mat. Sci. Eng. 2017, 245 (2), 022018. DOI: 10.1088/1757-899X/245/2/022018

3. Kiecana, M.; Kanty, P.; Luzynska, K. Optimal control time evaluation for dry DSM soil-cement composites. MATEC Web Conf. 2018, 251, 01023. DOI: 10.1051/matecconf/201825101023

Please use consequently MPa not MPA, CO2 not CO2 and generally try to unify notations (units)

Sincerely

Reviewer 3 Report

The authors would like to publish interesting article entitled "Review of Calcium-Based Binders in Soil Stabilization- Challenges - Problems-Calcined Clay as Partial Replacement to Produce Low-Carbon Cement". It is brand new, well-organized, well-written, and well-discussed. Moreover, as using cement and lime has become one of the main concerns for engineers and they negatively impact the environment and economy, prompting research about alternative material was very interested in every society. With a large amount of calcined clay used, the clinker content of cement can be lessened by as much as 50% when compared to traditional Portland cement system. For those reasons aforementioned, the journal can put the manuscript into current archive.

Round 2

Reviewer 2 Report

Dear Authors

I appreciate the work that you devoted to improve your interesting study

I do not have any detailed comments

Sincerely